Hairiness: the missing link between pollinators and pollination

Stavert Jamie R. jamie.stavert@gmail.com 1
Liñán-Cembrano Gustavo 2
Beggs Jacqueline R. 1
Howlett Brad G. 3
Pattemore David E. 4
Bartomeus Ignasi 5
1 Centre for Biodiversity and Biosecurity, School of Biological Sciences, The University of Auckland , Auckland , New Zealand
2 Instituto de Microelectrónica de Sevilla CSIC/Universidad de Sevilla , Sevilla , Spain
3 The New Zealand Institute for Plant & Food Research Limited , Christchurch , New Zealand
4 The New Zealand Institute for Plant & Food Research Limited , Hamilton , New Zealand
5 Integrative Ecology Department, Estación Biológica de Doñana (EBD-CSIC) , Sevilla , Spain
Cutler Chris
Electronic publication date: 2016 Dec 21
Publication date: 2016
Volume: 4
Electronic Location ID: e2779
Received 2016 Sep 8; Accepted 2016 Nov 2
Copyright: ©2016 Stavert et al.
Copyright year: 2016
Copyright holder: Stavert et al.
License: This is an open access article distributed under the terms of the Creative Commons Attribution License, which permits unrestricted use, distribution, reproduction and adaptation in any medium and for any purpose provided that it is properly attributed. For attribution, the original author(s), title, publication source (PeerJ) and either DOI or URL of the article must be cited.
License URL: https://creativecommons.org/licenses/by/4.0/

Keywords: Pollination, Pilosity, Entropy, Functional trait, Pollen deposition, Ecosystem function, Image analysis, Pollen load, SVD

Funding: BeeFun project PCIG14-GA-2013-631653 MBIE C11X1309 This research was supported by the University of Auckland, BeeFun project PCIG14-GA-2013-631653 and MBIE C11X1309 Bee Minus to Bee Plus and Beyond: Higher Yields From Smarter, Growth-focused Pollination Systems. The funders had no role in study design, data collection and analysis, decision to publish, or preparation of the manuscript.

==============================
Background

Functional traits are the primary biotic component driving organism influence on ecosystem functions; in consequence, traits are widely used in ecological research. However, most animal trait-based studies use easy-to-measure characteristics of species that are at best only weakly associated with functions. Animal-mediated pollination is a key ecosystem function and is likely to be influenced by pollinator traits, but to date no one has identified functional traits that are simple to measure and have good predictive power.

Methods

Here, we show that a simple, easy to measure trait (hairiness) can predict pollinator effectiveness with high accuracy. We used a novel image analysis method to calculate entropy values for insect body surfaces as a measure of hairiness. We evaluated the power of our method for predicting pollinator effectiveness by regressing pollinator hairiness (entropy) against single visit pollen deposition (SVD) and pollen loads on insects. We used linear models and AICC model selection to determine which body regions were the best predictors of SVD and pollen load.

Results

We found that hairiness can be used as a robust proxy of SVD. The best models for predicting SVD for the flower species Brassica rapa and Actinidia deliciosa were hairiness on the face and thorax as predictors (R2 = 0.98 and 0.91 respectively). The best model for predicting pollen load for B. rapa was hairiness on the face (R2 = 0.81).

Discussion

We suggest that the match between pollinator body region hairiness and plant reproductive structure morphology is a powerful predictor of pollinator effectiveness. We show that pollinator hairiness is strongly linked to pollination—an important ecosystem function, and provide a rigorous and time-efficient method for measuring hairiness. Identifying and accurately measuring key traits that drive ecosystem processes is critical as global change increasingly alters ecological communities, and subsequently, ecosystem functions worldwide.

Introduction

Trait-based approaches are now widely used in functional ecology, from the level of individual organisms to ecosystems (Cadotte, Carscadden & Mirotchnick, 2011). Functional traits are defined as the characteristics of an organism’s phenotype that determine its effect on ecosystem level processes (Naeem & Wright, 2003; Petchey & Gaston, 2006). Accordingly, functional traits are recognised as the primary biotic component by which organisms influence ecosystem functions (Gagic et al., 2015; Hillebrand & Matthiessen, 2009). Trait-based research is dominated by studies on plants and primary productivity, and little is known about key traits for animal-mediated and multi-trophic functions, particularly for terrestrial invertebrates (Didham, Leather & Basset, 2016; Gagic et al., 2015; Lavorel et al., 2013).

Most animal trait-based studies simply quantify easy-to-measure morphological characteristics, without a mechanistic underpinning to demonstrate these “traits” have any influence on the ecosystem function of interest (Didham, Leather & Basset, 2016). This results in low predictive power, particularly where trait selection lacks strong justification through explicit ecological questions (Gagic et al., 2015; Petchey & Gaston, 2006). If the ultimate goal of trait-based ecology is to identify the mechanisms that drive biodiversity impacts on ecosystem function, then traits must be quantifiable at the level of the individual organism, and be inherently linked to an ecosystem function (Bolnick et al., 2011; Pasari et al., 2013; Violle et al., 2007).

Methodology that allows collection of trait data in a rigorous yet time-efficient manner and with direct functional interpretation will greatly enhance the power of trait-based studies. Instead of subjectively selecting a large number of traits with unspecified links to ecosystem functions, it would be better to identify fewer, uncorrelated traits, that have a strong bearing on the function of interest (Carmona et al., 2016). Selecting traits that are measurable on a continuous scale, would also improve predictive power of studies (McGill et al., 2006; Violle et al., 2012). However, far greater time and effort is required to measure such traits, exacerbating the already demanding nature of trait-based community ecology (Petchey & Gaston, 2006).

Animal-mediated pollination is a multi-trophic function, driven by the interaction between animal pollinators and plants (Kremen et al., 2007). A majority of the world’s wild plant species are pollinated by animals (Ollerton, Winfree & Tarrant, 2011), and over a third of global crops are dependent on animal pollination (Klein et al., 2007). Understanding which pollinator traits determine the effectiveness of different pollinators is critical to understanding the mechanisms of pollination processes. However, current traits used in pollination studies often have weak associations with pollination function and/or have low predictive power. For example Larsen, Williams & Kremen (2005) used body mass to explain pollen deposition by solitary bees even when the relationship was weak and non-significant. Many trait-based pollination studies have subsequently used body mass or similar size measures, despite their low predictive power. Similarly, Hoehn et al. (2008) used spatial and temporal visitation preferences of bees to explain differences in plants reproductive output. They found significant relationships (i.e., low P values) between spatial and temporal visitation preferences and seed set, but with small R2 values, suggesting these traits have weak predictive power. To advance trait-based pollination research we require traits that are good predictors of pollination success.

Observational studies suggest that insect body hairs are important for collecting pollen that is used by insects for food and larval provisioning (Holloway, 1976; Thorp, 2000). Hairs facilitate active pollen collection, e.g., many bees have specialised hair structures called scopae that are used to transport pollen to the nest for larval provisioning (Thorp, 2000). Additionally, both bees and flies have hairs distributed across their body surfaces which act to passively collect pollen for adult feeding (Holloway, 1976). Differences in the density and distribution of hairs on pollen feeding insects likely reflects their feeding behaviour, the types of flowers they visit, and whether they use pollen for adult feeding and/or larval provisioning (Thorp, 2000). However, despite anecdotal evidence that insect body hairs are important for pollen collection and pollination, there is no proven method for measuring hairiness, nor is there evidence that hairier insects are more effective pollinators.

Here, we present a novel method based on image entropy analysis for quantifying pollinator hairiness. We define pollination effectiveness as single visit pollen deposition (SVD): the number of conspecific pollen grains deposited on a virgin stigma in a single visit (King, Ballantyne & Willmer, 2013; Ne’eman et al., 2010). SVD is a measure of an insects’ ability to acquire free pollen grains on the body surface and accurately deposit them on a conspecific stigma. We predict that hairiness, specifically on the body parts that contact the stigma, will have a strong association with SVD. We show that the best model for predicting pollinator SVD for pak choi Brassica rapa is highly predictive and includes hairiness of the face and thorax dorsal regions as predictors, and the face region alone explains more than 90% of the variation. Similarly, the best model for predicting SVD for kiwifruit Actinidia deliciosa includes the face and thorax ventral regions and has good predictive power. Our novel method for measuring hairiness is rigorous, time efficient and inherently linked to pollination function. Accordingly, this method could be applied in diverse trait-based pollination studies to progress understanding of the mechanisms that drive pollination processes.

Materials and Methods

Imaging for hairiness analysis

We photographed pinned insect specimens using the Visionary Digital Passport portable imaging system (Fig. 1). Images were taken with a Canon EOS 5D Mark II digital camera (5,616 × 3,744 pix). The camera colour profile was sRGB IEC61966-2.1, focal length was 65 mm and F-number was 4.5. We used ventral, dorsal and frontal shots with clear illumination to minimise reflection from shinny insect body surfaces. All photographs were taken on a plain white background. Raw images were exported to Helicon Focus 6 where they were stacked and stored in .jpg file format.

Figure 1 Entropy image of the face of a native New Zealand solitary bee Leioproctus paahaumaa (A) and the corresponding entropy image (B).

Warmer colours on the entropy image represent higher entropy values (shown by the scale bar on the right). Black dots on the entropy image are near-round and small objects that have been removed from the analysis by the pre-processing function.

Image processing and analysis

We produced code to quantify insect pollinator hairiness using MATLAB (MathWorks, Natick, MA, USA), and functions from the MATLAB Image Processing ToolBox. We quantified relative hairiness by creating an entropy image for each insect photograph, and computed the average entropy within user-defined regions (Gonzales, Woods & Eddins, 2004). To calculate entropy values for each image we designed three main functions. The first function allows the user to define up to four regions of interest (RoIs) within each image. The user can define regions by drawing contours as closed polygonal lines of any arbitrary number of vertexes. All information about regions (location, area and input image file name) is stored as a structure in a .mat file.

The second function executes image pre-processing. We found that some insects had pollen grains or other artefacts attached to their bodies, which would alter the entropy results. Our pre-processing function eliminates these objects from the image by running two filtering processes. First, the function eliminates small objects with an area less than the user definable threshold (8 pixels by default). For the first task, each marked region is segmented using an optimized threshold obtained by applying a spatially dependant thresholding technique. Once each region has been segmented, a labelling process is executed for all resulting objects and those with an area smaller than the minimum value defined by the user are removed. Secondly, as pollen grains are often round in shape, the function eliminates near-circular objects. The perimeter of each object is calculated and its similarity to a circle (S) id defined as: S=4π⋅AreaPerimeter2.

Objects with a similarity coefficient not within the bounds defined by the user (5% by default) are also removed from the image. Perimeter calculation is carried out by finding the object’s boundary, and computing the accumulated distance from pixel centre to pixel centre across the border, rather than simply counting the number of pixels in the border. The entropy filter will not process objects that have been marked as “deleted” by the pre-processing function. This initial pre-processing provides flexibility by allowing users to define the minimum area threshold and the degree of similarity of objects to a circle. Users can also disable the image pre-processing by toggling a flag when running the entropy filter.

Once pre-processing is complete, each image is passed to the third function, which is the entropy filter calculation stage. The entropy filter produces an overall measure of randomness within each of the user defined regions on the image. In information theory, entropy (also expressed as Shannon Entropy) is an indicator of the average amount of information contained in a message (Shannon, 1948). Therefore, Shannon Entropy, H, of a discrete random variable X that can take n possible values x1,x2,…,xn, with a probability mass function P(X) is given by: HX=−∑i=1nPxi⋅log2Pxi.

When this definition is used in image processing, local entropy defines the degree of complexity (variability) within a given neighbourhood around a pixel. In our case, this neighbourhood (often referred to as the structuring element) is a disk with radius r (we call the radius of influence) that can be defined by the user (7 pixels by default). Thus for a given pixel in position (i, j) in the input image, the entropy filter computes the histogram Gij (using 256 bins) of all pixels within its radius of influence, and returns its entropy value Hij as: Hij=−Gij⋅log2Gij,

where Gij is a vector containing the histogram results for pixel (i, j) and (⋅) is the dot product operator. Using default parameters, our entropy filter employs a 7 pixel (13 × 13 neighbourhood) radius of influence, and a disk-shaped structuring element, which we determined based on the size of hairs. Therefore, in the entropy image, each pixel takes a value of entropy when considering 160 pixels around it (by default). We determined the optimal radius of influence for the entropy filter by running our entropy function with the radius of influence set as a variable parameter. We then visually compared the contrast in areas of low vs. high hairiness in the resulting entropy images (i.e., Fig. 1). We found that a 7 pixel radius of influence gave the best contrast between low and high hairiness areas for our species set. Hair thickness values across species typically ranged between 3.5–4.5 pixels and therefore, the 7 pixel radius of influence is approximately two times the width of a hair.

The definition of the optimum radius of influence depends on the size of the morphological responsible for the complexity in the RoI. This is defined not only by the physical size of these features but also by the pixel-to-millimetre scaling factor (i.e., number of pixels in the sensor plane per mm in the scene plane). Thus, although 7 pixels is the optimum in our case to detect hairs, the entropy filter function takes this radius as an external parameter which can be adjusted by the user to meet their needs.

The entropy filter function is a process that runs over three different entropy layers (ER, EG, EB), one for each of the camera’s colour channels (Red, Green, and Blue), for each input image. These three images are combined into a final combined entropy image ES, where each pixel in position (i, j) takes the value ES(i, j): ESi,j=ERi,j⋅EGi,j⋅EGi,j

Once entropy calculations are complete, our function computes averages and standard deviations of ES within each of the regions previously defined by the user, and writes the results into a .csv file (one row per image). Entropy values produced by this function are consistent for different photos of the same region on the same specimen (Supplemental Information 5). The scripts for the image pre-processing, region marking and entropy analysis functions are provided, along with a MATLAB tutorial (Supplemental Information 1–4).

Hairiness as a predictor of SVD and pollen load

Model flower floral biology and pollinator collection

We used pak choi Brassica rapa var. chinensis (Brassicaceae) and kiwifruit Actinidia deliciosa (Actinidiaceae) as model flowers to determine if our measurement of insect hairiness is a good predictor of pollinator effectiveness.

Both B. rapa and A. deliciosa are important mass flowering global food crops (Klein et al., 2007; Rader et al., 2009). B. rapa has an actinomorphic open pollinated yellow flower with four sepals, four petals, and six stamens (four long and two short) (Walker, Kinzig & Langridge, 1999). The nectaries are located in the centre of the flower, between the stamens and the petals, forcing pollinators to introduce their head between the petals. B. rapa shows increased seed set in the presence of insect pollinators and the flowers are visited by a diverse assemblage of insects that differ in their ability to transfer pollen (Rader et al., 2013). A. deliciosa is dioecious with individual plants producing either male or female flowers. Flowers are large (4–6 cm in diameter) and typically have 5–9 white/cream coloured petals (Devi, Thakur & Garg, 2015). Flowers have multiple stamens and staminodes with yellow anthers. Female flowers have a large stigma with multiple branches that form a brush-like structure. Both male and female flowers do not produce nectar but both produce pollen, which acts as a reward to visitors. Like B. rapa, A. deliciosa flowers are visited by a diverse range of insects that differ in their ability to transfer pollen, and seed set is increased in the presence of insect pollinators (Craig et al., 1988).

We collected pollinating insects for image analysis during the summer of December 2014–January 2015. Insects were chilled immediately and then killed by freezing within 1 day and stored at −18 °C in individual vials. All insects were identified to species level with assistance from expert taxonomists.

Image processing

We measured the hairiness of 10 insect pollinator species (n = 8–10 individuals per species), across five families and two orders. This included social, semi-social and solitary bees and pollinating flies. Regions marked included: (1) face; (2) head dorsal; (3) head ventral; (4) front leg; (5) thorax dorsal; (6) thorax ventral; (7) abdomen dorsal and (8) abdomen ventral. All entropy analysis was carried out using our image processing method outlined above. For estimates of body size, we took multiple linear measurements (body length, body width, head length, head width, foreleg length and hind leg length) of each specimen using digital callipers and a dissecting microscope.

Single visit pollen deposition (SVD) and pollen load

For B. rapa we used SVD data for insect pollinators presented in Rader et al. (2009) and Howlett et al. (2011); a brief description of their methods follows.

Pollen deposition on stigmatic surfaces (SVD) was estimated using manipulation experiments. Virgin B. rapa inflorescences were bagged to exclude all pollinators. Once flowers had opened, the bag was removed, and flowers were observed until an insect visited and contacted the stigma in a single visit. The stigma was then removed and stored in gelatine-fuchsin and the insect was captured for later identification. SVD was quantified by counting all B. rapa pollen grains on the stigma. Mean values of SVD for each species are used in our regression models.

To quantify the number of pollen grains carried (pollen load), sensu Howlett et al. (2011), collected insects while foraging on B. rapa flowers. Insects were captured using plastic vials containing a rapid killing agent (ethyl acetate). Once dead, a cube of gelatine-fuchsin was used to remove all pollen from the insect’s body surface. Pollen collecting structures (e.g., corbiculae, scopae) were not included in analyses because pollen from these structures is not available for pollination. Slides were prepared in the field by melting the gelatine-fuchsin cubes containing pollen samples onto microscope slides. B. rapa pollen grains from each sample were then quantified by counting pollen grains in an equal-area subset from the sample and multiplying this by the number of equivalent sized subset areas within the total sample.

We measured SVD for A. deliciosa (n = 8–12 per pollinator species). SVD measurements were taken for insect movements from staminate to pistillate flowers, using a method that differed from B. rapa. Individual pistillate buds were enclosed within paper bags 2–3 days prior to opening, and were later used as test flowers to evaluate pollen deposition by flowering visiting species. Each bag was secured using a wire tie (coated in plastic) that was gently twisted to exclude pollinators from visiting the opening flowers. Following flower opening, the bag was removed and the flower pedicel abscised where it joined the vine. The test flower was then carefully positioned using forceps to hold the pedicel 1–2 cm from a staminate flower containing a foraging insect, avoiding any contacting between flowers. If the test flower was visited by an insect, we allowed it to forage with minimal disturbance until it moved from the flower on its own accord. The first stigma touched by the foraging insect was then lightly marked near its base using a fine black felt pen. We then placed the marked stigma onto a slide and applied a drop of Alexander stain (Dafni, 2007). Alexander stain was used due to its effectiveness to stain staminate and pistillate pollen differently (pistillate pollen—green-blue, staminate pollen—dark red) (Goodwin & Perry, 1992).

Statistical analyses

We used linear regression models and AICC (small sample corrected Akaike information criteria) model selection to determine if our measure of pollinator hairiness is a good predictor of SVD and pollen load. We constructed global models with SVD or pollen load as the response variable, body region as predictors and body length as an interaction i.e., SVD or pollen load ∼body length * entropy face + entropy head dorsal + entropy head ventral + front leg + entropy thorax dorsal + entropy thorax ventral + entropy abdomen dorsal + entropy abdomen ventral. We included body length in our global model as a proxy for body size as it had high correlation coefficients (Pearson’s r > 0.7) with all other body size measurements. Global linear models were constructed using the lm(stats) function. AICC model selection was carried out on the global models using the function glmulti() with fitfunction = “lm” in the package glmulti. We examined heteroscedasticity and normality of errors of models by visually inspecting diagnostic plots using the glmulti package (Crawley, 2002). Variance inflation factors (VIF) of predictor variables were checked for the best models using the vif() function in the car package. All analyses were done in R version 3.2.4 (R Core Team, 2014).

Results

Body hairiness as a predictor of SVD

For SVD on B. rapa, the face and thorax dorsal regions were retained in the best model selected by AICC, which had an adjusted R2 value of 0.98. The subsequent top models within 10 AICC points all retained the face and thorax dorsal regions and additionally included the abdomen ventral (adjusted R2 = 0.98), head dorsal (adjusted R2 = 0.98), and thorax ventral (adjusted R2 = 0.97) and front leg (adjusted R2 = 0.97) regions respectively (Table 1; Fig. 2). The model with the face region included as a single predictor had an adjusted R2 value of 0.88, indicating that this region alone explained a majority of the variation in the top SVD models.

Table 1 Regression models examining the effect of entropy on SVD and pollen load.

Top regression models examining the effect of insect body region entropy on single visit pollen deposition (SVD) for Brassica rapa and Actinidia deliciosa and pollen load for B. rapa. Models are presented in ascending order based on AICC values. Top models for each response variable are highlighted in bold.

Response variable	Model	Adj R2	AICc	Δi	wi	acc wi	
SVD (B. rapa)	Face+ Thorax dorsal	0.98	88.29	0.00	0.82	0.82	
Face + Thorax dorsal + Abdomen ventral	0.98	93.09	4.80	0.07	0.89	
Face + Head dorsal + Thorax dorsal	0.98	93.81	5.52	0.05	0.94	
Face + Thorax ventral + Thorax dorsal	0.97	96.59	8.29	0.01	0.96	
Face + Thorax dorsal + Front leg	0.97	97.02	8.72	0.01	0.97	
Pollen load (B. rapa)	Face	0.81	168.47	0.00	0.64	0.64	
Abdomen dorsal	0.73	171.59	3.12	0.13	0.78	
Face + Head dorsal	0.83	173.59	5.12	0.05	0.83	
Face + Abdomen dorsal	0.82	173.76	5.29	0.05	0.87	
Abdomen dorsal + Front leg	0.80	174.86	6.39	0.03	0.90	
SVD (A. deliciosa)	Face+ Thorax ventral	0.91	74.18	0.00	0.15	0.15	
Abdomen dorsal	0.81	74.21	0.03	0.15	0.30	
Face	0.80	74.35	0.17	0.14	0.45	
Head ventral	0.79	74.84	0.66	0.11	0.56	
Abdomen ventral	0.78	75.08	0.90	0.10	0.65	
Notes.

Δi is the difference in the AICC value of each model compared with the AICC value for the top model. wi is the Akaike weight for each model and acc wi is the cumulative Akaike weight.

Figure 2 Relationships between mean entropy for each body region and mean single visit pollen deposition (SVD) on Brassica rapa for 10 different insect pollinator species.

Black lines are regressions for simple linear models.

The best model for predicting SVD on A. deliciosa included the face and thorax ventral regions as predictors (adjusted R2 = 0.91) (Table 1; Fig. 3). However, the subsequent top four models were within two AICC points of the best model and therefore cannot be discounted as the potential top model. The face, thorax ventral, head ventral and abdomen ventral regions were retained in four of the five top models, which indicates that hairiness of the face and ventral regions is important for pollen deposition on A. deliciosa. For both B. rapa and A. deliciosa, body length and the body length interaction were not included in the top models.

Figure 3 Relationships between mean entropy for each body region and mean single visit pollen deposition (SVD) on Actinidia deliciosafor 7 different insect pollinator species.

Black lines are regressions for simple linear models.

Body hairiness as a predictor of pollen load

The best model for pollen load retained the face region only and had an adjusted R2 value of 0.81 (Fig. 4; Table 1). The subsequent best models retained the abdomen dorsal (adjusted R2 value of 0.73), the face and head dorsal (adjusted R2 = 0.83), the face and abdomen dorsal (adjusted R2 = 0.82) and the abdomen dorsal and front leg (adjusted R2 = 0.8) regions respectively. For pollen load, body length and the body length interaction were not included in the top models.

Figure 4 Relationship between entropy and Brassica rapa pollen load on insects.

Relationships between mean entropy for each body region and the mean number of Brassica rapa pollen grains carried by 9 different insect pollinator species. Black lines are regressions for simple linear models.

Discussion

Here we present a rigorous and time-efficient method for quantifying hairiness, and demonstrate that this measure is an important pollinator functional trait. We show that insect pollinator hairiness is a strong predictor of SVD for the open-pollinated flower B. rapa. Linear models that included multiple body regions as predictors had the highest predictive power; the top model for SVD retained the face and thorax dorsal regions. However, the face region was retained in all of the top models, and when included as a single predictor, had a very strong positive association with SVD. In addition, we show that hairiness, particularly on the face and ventral regions, is a good predictor of SVD for A. deliciosa, which has a different floral morphology, suggesting our method could be suitable for a range of flower types. Hairiness was also a good predictor for pollen load, and the face region was again retained in the top model for B. rapa. The abdomen dorsal, head dorsal and front leg regions were also good predictors of pollen load and were retained in the subsequent top models. Our results validate the importance of insect body hairs for transporting and depositing pollen. Surprisingly, we did not find strong associations between SVD and body size, and top models did not contain the body length interaction. Similarly, body length was not retained in the top models for pollen load. This indicates that our measure of hairiness has far greater predictive power than body size for both SVD and pollen load.

When deciding on which body regions to measure hairiness, researchers may first need to assess additional pollinator traits, such as flower visiting behaviour. This is because the way in which insects interact with flowers influences what body parts most frequently contact the floral reproductive structures (Roubik, 2000). For some open pollinated flowers, such as B. rapa, facial hairs are probably the most important for pollen deposition because the face is the most likely region to contact the anthers and stigma. However, for flowers with different floral morphologies, facial hairs may not be as important because the floral reproductive structures have different positions relative to the insect’s body structures. For example, disc-shaped flowers tend to deposit their pollen on the ventral regions of pollinators, while labiate flowers deposit their pollen on the dorsal regions (Bartomeus, Bosch & Vilà, 2008). We found that hairiness on the face and ventral regions of pollinators was most important for pollen deposition on A. deliciosa flowers. The reproductive parts of A. deliciosa form a brush shaped structure and therefore are most likely to contact the face and ventral surfaces of pollinators. Accordingly, where studies focus on a single plant species i.e., crop based studies, it is important to consider trait matching when selecting pollinator body region(s) to analyse (Butterfield & Suding, 2013; Garibaldi et al., 2015).

It is important to consider that pollinator performance is a function of both SVD and visitation frequency, and these two components operate independently (Kremen, Williams & Thorp, 2002; Mayfield, Waser & Price, 2001). Here, we focus on a single trait that is important for pollinator efficiency (SVD), but to calculate pollinator performance researchers need to measure both efficiency and visitation rate. Additional pollinator traits related to visitation rate, as well as other behavioural traits such as activity patterns relative to the timing of stigma receptivity (Potts, Dafni & Ne’eman, 2001) and foraging behaviour, e.g., nectar vs. pollen foraging (Herrera, 1987; Javorek, Mackenzie & Kloet, 2002; Rathcke, 1983), may be important for predicting pollination performance. In some circumstances it might also be important to consider trait differences between male and female pollinators, particularly for some bee species. Male and female bees may have different pollen deposition efficiency due to differences in their foraging behaviour and resource requirements. For example, female bees are likely to visit flowers to collect pollen for nest provisioning while males simply consume nectar and pollen during visits (Cane, Sampson & Miller, 2011). For some flowers, male bees have a similar pollination efficiency compared to females (e.g., summer squash Cucurbita pepo; Cane, Sampson & Miller, 2011) while for others, female bees are more effective than males (e.g., lowbush blueberry Vaccinium angustifolium; Javorek, Mackenzie & Kloet, 2002).

For community-level studies that use functional diversity approaches, our method could be used to quantify hairiness for several body regions and weighted to give better representation of trait diversity within the pollinator community. This is necessary where plant communities contain diverse floral traits i.e., open-pollinated vs. closed-tubular flowers (Fontaine et al., 2006). Hairs on different areas of the insect body are likely to vary in relative importance for pollen deposition depending on trait matching (Bartomeus et al., 2016). Our method requires hairiness to be measured at the individual-level (Fig. S1), which makes it an ideal trait to use in new functional diversity frameworks that use trait probabilistic densities rather than trait averages (Carmona et al., 2016; Fontana et al., 2016). Combining predictive traits, such as pollinator hairiness, with new methods that amalgamate intraspecific trait variation with multidimensional functional diversity, will greatly improve the explanatory power of trait-based pollination studies.

One of the greatest constraints to advancing trait-based ecology is the time-demanding nature of collecting trait data. This is because ecological communities typically contain many species, which have multiple traits that need to be measured and replicated (Petchey & Gaston, 2006). To improve the predictive power of trait-based ecology and streamline the data collection process we must firstly identify traits that are strongly linked to ecosystem functions and secondly, develop rigorous and time-efficient methodologies to measure traits at the individual level. We achieve this by providing a method for quantifying a highly predictive trait at the individual-level, in a time-efficient manner. Our method also complements other recently developed predictive methods for estimating difficult-to-measure traits that are important for pollination processes i.e., bee tongue length; Cariveau et al. (2016).

Predicating the functional importance of organisms is critical in a rapidly changing environment where accelerating biodiversity loss threatens ecosystem functions (McGill et al., 2015). Our novel method for measuring pollinator hairiness could be used in any studies that require quantification of hairiness, such as understanding adhesion in insects (Bullock, Drechsler & Federle, 2008; Clemente et al., 2010) or epizoochory (Albert et al., 2015; Sorensen, 1986). It is also a much needed addition to the pollination biologist’s toolbox, and will progress the endeavour to standardise trait-based approaches in pollination research. This is a crucial step towards developing a strong mechanistic underpinning for trait-based pollination research.

Supplemental Information

Supplemental Information 1 Pre-process region script

Click here for additional data file.

Supplemental Information 2 Mark regions script

Click here for additional data file.

Supplemental Information 3 Entropy test script

Click here for additional data file.

Supplemental Information 4 MATLAB tutorial

Click here for additional data file.

Supplemental Information 5 Variation in entropy values between different photos of the same specimen

Variation in entropy values for multiple photos (n = 5 per region per species) for two different body regions of the same specimen.

Click here for additional data file.

Supplemental Information 6 Entropy and SVD dataset

This dataset includes mean entropy values for 8 body regions and body length measurements for 10 pollinator species. It also includes mean SVD values for Brassica rapaand Actinidia deliciosaand mean pollen load values for Brassica rapa.

Click here for additional data file.

Figure S1 Intraspecific variation in pollinator hairiness

Intraspecific variation in entropy values across different body regions of insect pollinators used in our study. Boxes represent the interquartile range, horizontal lines within boxes are median values, whiskers are the range and single dots are outliers.

Click here for additional data file.

Figure S2 Brassica rapa flower morphology

Photograph of a pak choi Brassica rapa flower. Labels show the key reproductive structures.

Click here for additional data file.

Figure S3 Actinidia deliciosa flower morphology

Photographs of a female (a) and male (b) kiwifruit Actinidia deliciosa flower. Labels show the key reproductive structures.

Click here for additional data file.

We thank David Seldon, Adrian Turner and Iain McDonald for assistance photographing insect specimens, Anna Kokeny for help collecting specimens and Stephen Thorpe for assistance identifying specimens. Patrick Garvey and Greg Holwell provided insightful comments on the earlier manuscript. We also thank two anonymous reviewers for helpful and constructive comments on the manuscript. We thank Sam Read, Brian Cutting, Heather McBrydie, Alex Benoist, Rachel L’helgoualc’h and Simon Cornut for assistance in field work. Lastly, we thank Estación Biológica de Doñana for hosting JS while developing the methodology for this paper.

Additional Information and Declarations

Competing Interests

Author Contributions

Data Availability

Brad G. Howlett and David E. Pattemore are employees of The New Zealand Institute for Plant & Food Research Limited. All other authors have no competing interests.

Jamie R. Stavert conceived and designed the experiments, performed the experiments, analyzed the data, contributed reagents/materials/analysis tools, wrote the paper, prepared figures and/or tables, reviewed drafts of the paper.

Gustavo Liñán-Cembrano and Ignasi Bartomeus conceived and designed the experiments, analyzed the data, contributed reagents/materials/analysis tools, wrote the paper, reviewed drafts of the paper.

Jacqueline R. Beggs conceived and designed the experiments, wrote the paper, reviewed drafts of the paper.

Brad G. Howlett conceived and designed the experiments, performed the experiments, reviewed drafts of the paper.

David E. Pattemore conceived and designed the experiments, performed the experiments.

The following information was supplied regarding data availability:

The raw data has been supplied as a Supplemental File.

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
