# Peer review of "Hairiness: the missing link between pollinators and pollination"

_PeerJ, doi:10.7717/peerj.2779_

## Round 0.1 · original submission · Minor Revisions

I want to apologize to the authors for the delay in coming to a decision on their paper. Seven other invited referees declined or did not respond to invitations to review. Fortunately, two expert referees have now provided insightful and thorough reviews of what I think is a well-constructed, well-written, interesting, innovative, and important contribution. My recommendation is that the paper needs minor revisions, but that the points raised by the referees should be able to be addressed by the authors without too much difficulty. In particular, I would like to see the following:

- Add discussion to acknowledge that there may well be situations where hairiness is not a good proxy for pollination efficiency. R1 provided excellent commentary and citations to consider.
- Add abstract headings (R2)
- Consider adding as supplementary material, as suggested by R2

Reviewer 1 ·

Basic reporting

No Comments

Experimental design

This is fine, though as stated below, I am not sure if the single plant/small group of pollinators used here is indicative of the range of plant-pollinator interactions.

Validity of the findings

No Comments

Additional comments

Greetings,

I have now finished my review of the manuscript “Hairiness: the missing link between pollinators and pollination“. In general, I find this manuscript to be very well written, very interesting, and of great value to those interested in functional traits in plant-pollinator interaction studies. I have no grammatical or other editorial comments on the manuscript. I think the technique developed and utilized in the manuscript will be of great use in pollination biology, and think it should be published.

My only concern is that I feel the authors are simplifying things too much – only one crop with a limited range of pollinators are used as a case study for proof of concept; this system, and the likely range of floral visitors, are not representative of most crops. For instance, in the supplemental materials, they mention two relatively hairy bee species, Bombus terrestris, a species of Leioproctus, and a small group of other non-bee pollinators. The authors claim that the “hairiness” level of a “species” is useful trait, and offer means to evaluate it, though in the absence of comparative pollinator behavioral data, some may argue that hairiness alone may not be enough in some plant-pollinator studies (they do briefly address behavior on lines 287-288 of the manuscript, indicating that “[w]hen deciding on which body regions to measure hairiness, researchers may first need to assess additional pollinator traits, such as flower visiting behaviour” which is of course very true, though I feel this warrants more discussion).

For instance, based on hairiness alone, some cleptoparasitic bee genera (e.g., Coelioxys, many melectine bees), male bumble bees and males of many other bee species should be as good, or even better pollinators than conspecific females, though in general, they are not considered to be so (but very few of these taxa have been/are studied). Also, some flies are obviously more hairy than bees, but I would suspect that a pollen collecting bee would potentially have more pollen on its body than a fly of comparable hairness, and more than a nectar collecting bee of comparable hairiness. So in reference to the present study, are all individual Bombus terrestris visitors, and all Leioproctus visitors comparable to each other, or would the authors expect differences among individuals of one species in SVD based on sex, nectar versus pollen foraging (e.g., males, cleptos), etc.? The differences between what male and female bees pick up/deposit on stigmas are largely due to their behaviors on flowers (i.e., nectar feeding only, looking for females, sleeping) (see Javorek et al. 2002). I list several studies below where male bees are evaluated or indicated as pollinators; Cane et al.’s study (Cane et al. 2010) offered one assessment of male bees, indicating that adequate pollination from males was obtained for the target crop (summer squash), but also suggested (Cane 2002; cited in that study) that male solitary bees in general may do a reasonable job based on visits, especially if they are oligolectic species, but in general visit fewer flowers than their females. The paper the authors cite by Thorp (Thorp 2000) also has some information about male bees, but the authors make no mention of males versus females in terms of comparable pollination ability. In general, these should not be ignored, as many flower visiting insects, even hairy ones, are visiting for nectar, not pollen, and based on the behavior on the flower, pollen “pick up” and subsequent pollination could be influenced by factors other than hairiness. The study by Javorek et al. (2002) offers some comparison of male and female bees in lowbush blueberry, and differences were observed in females of a species depending on whether or not they were collecting pollen versus nectar.

So, in summary, I think this is an excellent paper and recommend it for publication, but suggest the authors add more discussion to help explain conditions in which hairiness alone may not be the missing link. Should researchers wanting to utilize this interesting and valuable approach evaluate for differences in males and females of bees, and based on behavior?

Cane, J.H., Sampson, B.J. and Miller, S.A., 2011. Pollination value of male bees: the specialist bee Peponapis pruinosa (Apidae) at summer squash (Cucurbita pepo). Environmental Entomology, 40(3): 614-620.

Dafni, A., Ivri, Y. and Brantjes, N.B.M., 1981. Pollination of Serapias vomeracea Briq.(Orchidaceae) by imitation of holes for sleeping solitary male bees (Hymenoptera). Acta Botanica Neerlandica, 30(1-2): 69-73.

Javorek, S.K., Mackenzie, K.E. and Vander Kloet, S.P., 2002. Comparative pollination effectiveness among bees (Hymenoptera: Apoidea) on lowbush blueberry (Ericaceae: Vaccinium angustifolium). Annals of the Entomological Society of America, 95(3): 345-351.

Sapir, Y., Shmida, A. and Ne'eman, G., 2005. Pollination of Oncocyclus irises (Iris: Iridaceae) by night-sheltering male bees. Plant Biology, 7(04): 417-424.

Reviewer 2 ·

Basic reporting

The manuscript is well written in a clear language. All basic requirements listed by PeerJ criteria are met, however, some general and specific points for adjustments and suggestions are made below.

The Abstract is not in the PeerJ standard and an adjustment should be made, which is the addition of the headings: Background, Methods, Results and Discussion (https://peerj.com/about/author-instructions/#standard-sections). The text has been constructed as such and alteration should take place only to fit the journal’s standards unless chief-editor feels unnecessary.

On Line 189 of the pdf: This manuscript should not be considered a Case Study, as it presents a sound hypothesis, being tested and further discussed. Aims and Scope say that Case Studies are not suitable for publication in PeerJ, and should be submitted to PeerJ Preprint. Therefore, I recommend that the sub-section title should be changed by simply deleting the “Case study:” expression.

On Line 194 of the pdf: I would like to suggest that a picture of the flower or a schematic is provided in the manuscript or Supplemental material. This will certainly fully illustrate the publication and allow it to have an even broader readership.

On Line 207 of the pdf: This section title should be deleted and text be merged with the previous one, otherwise it get’s a bit confusing since there is an “Image processing” section above.

On Line 225 of the pdf: Put in parenthesis or within commas: ", sensu Howlett et al. (2011),”, the way that is written is a bit confusing.

Experimental design

The authors describe and validate an image processing method that could be used to predict pollination success by measuring the amount of hair (or hairiness) an insect has throughout different regions of its body. They use Entropy as a measurement for hairiness, which is a good source of information about the roughness and texture of a surface (e.g. face). Therefore, the greater the entropy, the greater the insect hairiness is; and thus, the greater the probability of the insect to load and deposit grains of pollens while visiting flowers. They have presented a sound hypothesis and good experimental design as a basis for their results and discussion. The methods are well described and give enough information for another researcher to reproduce or use the same techniques.

One suggestion that I make is that, on Line 169 of the pdf, the authors should give an example of single hair measurement values (in pixels), so the reader can relate to the disk element used for analysis; and, if need be, the reader will be able to understand why and which changes in settings for the values for more or less neighbouring pixels is important in their own research.

Validity of the findings

I find the data sufficiently robust to defend their results and discussion. However, I understand that, on Line 282 of the pdf, the authors discuss that body length and body size did not relate to the SVD or pollen loading. This is an important statement and should be consolidated by presenting the results in the Supplemental material, as it is not primary for the main findings, but it is still an interesting result for their discussion and robustness of the method relative to what has been used so far.

Moreover, I find the results for the kiwi flower very interesting and, if incorporated into the main manuscript, it could be discussed comparatively with figures side by side. This would demonstrate more clearly the success of the methodology within the manuscript. The reader wound not need to go to the Supplemental material to grasp the idea of how widely applicable is the method, which would only facilitate the reader to understand how the frame-work can be used in different situations (already pointed out by the authors in the discussion). Nonetheless, not incorporating the kiwi results into the manuscript not downgrade the present manuscript, and should be considered as a suggestion of improvement.

Additional comments

Some of my suggestions will only enrich and improve the manuscript and would be up to the editor to decide if a re-review is necessary. In my point of view the manuscript need some minor adjustments only, and, overall, the research is well designed and presented good results. I recommend the publication in PeerJ.

Annotated reviews are not available for download in order to protect the identity of reviewers who chose to remain anonymous.

---

## Round 0.2 · Minor Revisions

In the annotated manuscript I have included a few minor edits and comments, and a comment regarding the mention of body size/length (it seems only length was really considered) in the M&M and Results, as it relates to statements in the Discussion.

---

## Round 0.3 · accepted · Accept

The most recent minor changes you made to the manuscript have addressed all my concerns.